# Neuropsychological Disability in the Case of Natalizumab-Related Progressive Multifocal Leukoencephalopathy

**DOI:** 10.3390/medicina58040551

**Published:** 2022-04-17

**Authors:** Viviana Lo Buono, Giangaetano D’Aleo, Simona Cammaroto, Maria Cristina De Cola, Francesca Palmese, Chiara Smorto, Silvia Marino, Giuseppe Venuti, Edoardo Sessa, Carmela Rifici, Francesco Corallo

**Affiliations:** 1IRCCS Centro Neurolesi Bonino Pulejo, 98100 Messina, Italy; viv.lobuono@gmail.com (V.L.B.); giangaetano.daleo@irccsme.it (G.D.); simona.cammaroto@irccsme.it (S.C.); chiara.smorto@irccsme.it (C.S.); silvia.marino@irccsme.it (S.M.); giuseppe.venuti@irccsme.it (G.V.); edoardo.sessa@irccsme.it (E.S.); carmela.rifici@irccsme.it (C.R.); francesco.corallo@irccsme.it (F.C.); 2Azienda ULSS Marca Trevigiana, Ospedale Cà Foncello, 31100 Treviso, Italy; francesca.palmese@gmail.com

**Keywords:** cognitive impairment, multiple sclerosis, progressive multifocal leukoencephalopathy

## Abstract

Background: Progressive multifocal leukoencephalopathy (PML) is a viral disease characterized by progressive damage or inflammation of the cerebral white matter that can be encountered in patients with multiple sclerosis (MS). There are cases of PML caused by pharmacological agents including natalizumab. Therefore, in patients treated with this drug, early identification of PML allows changes in the treatment plan, reducing the risks of morbidity and mortality. Case presentation: We reported the case of a 57-year-old female diagnosed with relapsing-remitting MS, who presented with PML related to natalizumab. The patient presented with change in behavioral, radiological abnormalities in the left parieto-temporal lobes. We described the longitudinal course of PML, from the diagnosis until the patient’s death, documenting the progressive deterioration of her cognitive functioning, supported by changes on sequential brain scans and neurophysiological data. Conclusion: The neuropsychological impairment documented in this case study expands the range of treatment-related complications associated with natalizumab, and provides evidence that occurrence of “atypical” cognitive deficits in MS may support the early diagnosis of PML.

## 1. Introduction

Progressive multifocal leukoencephalopathy (PML) is a rare and destructive demyelinating disease of the central nervous system (CNS) caused by the John Cunningham virus (JCV) and, in almost all instances, has been associated with a significant underlying abnormality in cell-mediated immunity [1]. In 2005, PML was described for the first time in patients with multiple sclerosis (MS) taking natalizumab and interferon beta-1 [2]. Natalizumab is a humanized monoclonal antibody against integrin beta-4 with highly effective therapy for relapsing-remitting MS. A few cases of PML in MS patients have been described in individuals treated with other agents including dimethyl fumarate and fingolimod [3] but with a much lower prevalence than that associated with natalizumab [4]. 

The distinguishing features of the PML are progressive focal deficits such as visual, motor and vision dysfunction resulting from brain lesions [5]. The diagnosis of PML is confirmed either by brain biopsy revealing typical neuropathological features or by detection of JCV DNA in cerebrospinal fluid [6]. However, due to the relative lack of specificity in neurological symptoms related to PML, the occurrence of new focal deficits may be misinterpreted as a MS relapse, resulting in delay in diagnosis [7]. 

Neuropsychological impairments can occur at the onset of PML [8]. Behavioral and speech symptoms have also been described in PML during the course of disease [9]. It would appear that these neuropsychological dysfunctions can be distinguished from MS-related cognitive decline and might therefore support early diagnosis; however, the prevalence of specific cognitive disorders as distinctive signs of PML has not yet been clarified and is still under investigation [10].

We reported the clinical course of natalizumab-related PML (NTZ-PML) in a woman with relapsing-remitting MS. We aimed to identify possible cognitive peculiarities or atypical symptoms in the first signs of PML. 

## 2. Case Presentation

A 57-year-old woman was diagnosed with relapsing-remitting MS in 2005. Initial treatment with interferon beta-1b (Extavia^®^, Betaferon^®^) was switched to natalizumab in 2017 due to persistent disease activity. History does not report whether the patient was on immunosuppressive drug therapy. The patient started treatment with monthly intravenous infusions of 300 mg natalizumab, following informed consent and according to the European Medicine Agency (www.ema.europa.eu/en/about-us/contact/how-find-us accessed on 26 May 2016). Informed consent was obtained, at the time of the first appointment, at the outpatient clinic and the purposes were both therapeutic and scientific.

Subsequently, no clinical or MRI signs of new MS activity were evident until July 2018 (T0). The patient was routinely followed up with clinical evaluations and MRI every three months.

She was clinically stable, and at the Kurtzke Expanded Disability Status Scale (EDSS) she achieved 1. This scale is specific to measure disability in multiple sclerosis [11] by assigning a score in different functional areas (ranging from 0 = normal to 10 = death due to MS) such as pyramidal, cerebellar brainstem, sensory bowel and bladder, visual, cerebral, and others.

She was working and well-integrated in the social context and had a good cognitive performance in all examined domains (see Table 1).

In September 2018, the patient had shown behavioral alterations, such as mild disinhibition, irritability and aggression, and episodic spatial disorientation. MRI showed lesions suspicious for PML near the corpus callosum and massive extension of the area of altered signal intensity in the left parieto-temporal lobes (Figure 1). Natalizumab was immediately discontinued. The patient had to discontinue infusion therapy via natalizumab after 42 administrations, and begun to took mirtazapine therapy (30 mg). She was admitted to hospital in order to monitor the course of the disease. The day after hospital admission, she underwent cerebrospinal fluid test for JCV. Indeed, calculation anti-JCV antibody index allows further PML risk stratification [12].

The cerebrospinal fluid tested by Unilabs a.s. (Copenhagen, Denmark) revealed the presence of JCV DNA (quantitative copies/mL 1,087,000). Thus, anti-JCV-serum-antibodies were positive with an index value of 4.39 (substantially above the 1.5 threshold), indicating a high risk for PML. 

A complication of PML treatment is the development of immune reconstitution inflammatory syndrome (IRIS). Caused by the transition from acquired immunodeficiency to a state of immune reconstitution, it leads to excessive cytotoxic damage after activation of T lymphocytes upon contact with JCV-infected cells [13]. Although IRIS can appear weeks to months after discontinuation of the drug, the patient did not suffer from PML-IRIS. Neuropsychological assessment was conducted by the clinical psychologist. At admission to hospital (T1) the patient was alert and cooperative, but she presented with an acute state of confusion. She showed nominal aphasia. The informative content of speech was valid, although at times tending towards confabulation. Verbal comprehension was adequate. The neuropsychological assessment revealed cognitive impairment in several domains: attention deficit, especially in tasks requiring greater concentration, associated with a slowing of the speed of information processing; deficit of memory skills, both in the coding of short-term information and in the retrieval of long-term information; deficit in visual-spatial coordination skills. The ability for analysis and deductive logical reasoning was preserved, as well as insight. Anxiety-depressive mood alteration was observed. EEG showed altered brain electrical activity due to the presence of theta activity mixed with medium-to-large voltage slow waves on the left anterior leads.

At the clinical re-evaluation performed, after 30 days (T2), the patient presented a clinical and neuropsychological worsening. The patient appeared to be alert and cooperative; she carried out simple orders and answered questions appropriately. Conversation was possible with the help of the interlocutor, and spontaneous speech was impaired by aphasic deficits. Oral and written lexico-semantic production was characterized by deficits in access and phonological structuring.

There were many anomies, phonemic and rare semantic paraphasia. To retrieve the lexical information, the patient needed help, and frequent phonological errors of inversion and/or substitution in word structure were found. Although oral and written comprehension was preserved, during reading she required long latency times to retrieve the information and showed access to it through the spontaneous use of compensatory strategies. The patient had limitations, requiring assistance and supervision in basic activities of daily living (EDSS: 8.5). MRI showed worsening of radiological abnormalities with new involvement of the frontal lobe (Figure 2).

After about four weeks (T3), at the follow-up examination, carried out concomitantly with the worsening of the clinical situation, the patient was soporific and not contactable. MRI showed expansion of the bilateral damage and frontal progression (Figure 3).

The patient’s clinical condition worsened further and she was transferred to the intensive care unit. EEG examination showed the presence of diffuse low-voltage theta/delta activity. 

The course of the disease was progressive and, despite the suspension of drug treatment, the patient died about a month later because of widespread CNS inflammation.

## 3. Discussion

This case described clinical feature and neuropsychological dysfunctions during different phases of the NTZ-PML disease course using a battery of standardized tests, in order to identify specific early cognitive symptoms that could support the diagnosis. Clinical and radiological feature at PML onset has been well described in the literature [14], while the neuropsychological features have often been underestimated.

Cognitive dysfunctions are frequent and disabling symptoms in MS, and typically manifest as a general cognitive slowdown that involves information processing speed, attention, working memory and executive functions [15,16,17,18]. Previous studies have shown that cognitive deficits at PML onset are uncommon in MS patients, such as aphasia and disorientation, and therefore they should receive adequate clinician attention [10]. Clifford et al. [19] described speech deficit at PML onset related to left frontal subcortical white matter lesions. In the study by Dahlhaus et al. [20], depression and cognitive changes indicated the development of PML. 

In this reported clinical case, behavioral alterations, such as disinhibition and irritability, were the first strange and uncommon symptoms of MS that prompted the patient’s family to request a new outpatient visit. Other cognitive dysfunctions present at the onset of PML were spatial disorientation and aphasia [21]. The patient also showed anxious and depressive symptoms. In the subsequent neuropsychological evaluations, we found a marked decline in working memory and attentional processes, and progressively we registered a decline in reasoning, problem solving and executive functioning. Similarly, to the study by Kinner et al. [17], a rapid decline in all cognitive domains was found. 

Simultaneously to the neuropsychological worsening, the serial MRI showed the rapid evolution of the inflammatory infectious pathological process of PML, which never presented areas of impregnation in the context, prognostically negative sign reported in the literature and expression of a lack of response of the patient’s immune system.

## 4. Conclusions

PML is one of the most serious treatment-related complications encountered in MS patients [22]. Although selected drug therapies for MS such as natalizumab reduce the risk of disability progression, patients should be adequately informed about their risk of developing PML before starting treatment, so that they can make an informed choice. In addition, they should be periodically evaluated for PML risk [23]. To date, there is no specific and effective treatment for PML; the infection outcome depends on the individual’s immune reconstitution ability to respond to JCV [24]. The prognosis of PML is often poor, since it is associated with a significant risk of long-term disability and mortality. The goal of treatment of natalizumab-associated PML is the restoration of immune function through rapid removal of the drug [25,26]. 

Neuropsychological evaluation in MS patients taking natalizumab should be performed regularly, with tests measuring all cognitive domains: attention, information processing speed, memory, executive functions, language, visual-spatial abilities, and behavioral alterations. Its repetition during follow-up would be able to monitor the evolution of these deficits over time. Early identification of cognitive deficits suggestive of PML could support the neurologist to diagnosis and withdrawing of natalizumab for a better prognosis. 

## Figures and Tables

**Figure 1 medicina-58-00551-f001:**
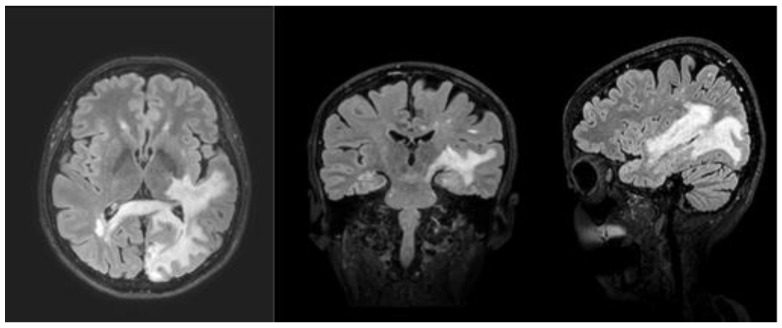
FLAIR3D showed massive area of altered signal intensity in the left occipital lobe and in the anterior ipsilateral temporal lobe, with initial involvement of the ipsilateral posterior nucleus-capsular region.

**Figure 2 medicina-58-00551-f002:**
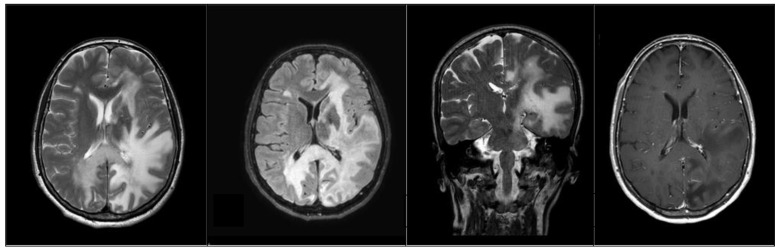
Progression of the area of altered signal intensity that extends to the left frontal lobe and passes the midline posteriorly through the corpus callosum into the right parietal and occipital lobe. FLAIR image showed the front of the infectious inflammatory pathological process, in the deep temporal region. There are no areas of pathological enhancement.

**Figure 3 medicina-58-00551-f003:**
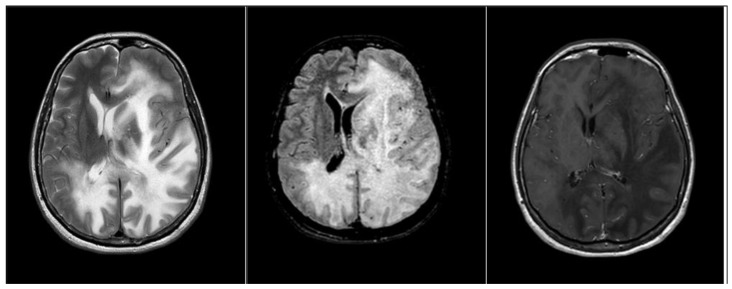
The signal alteration involved both the cerebellar lobes and the structures of the brain stem in the subtentorial site, and the left hemisphere with extension to the capsule nucleus region, causing a significant mass effect on the ipsilateral ventricular hemisystem and consequent shift of the structures of the midline. The pathological process, which also extends contralaterally in the right hemisphere, does not present areas of pathological enhancement in the context.

**Table 1 medicina-58-00551-t001:** Comparison of neuropsychological score, before PML related-natalizumab diagnosis (T0) at the onset of the first symptoms of PML (T1), 30 days after hospitalization (T2) and 60 days after hospitalization (T3). The cut-offs refer to the Italian calibration of the tests used.

Subtest	T0	T1	T2	T3	Cut-Off
MMSE	30/30	26/30	13/30	----	26
FAB	18/18	13/18	6/18	----	14
SRT-LTS	64.15	14.96	1.96	----	23.3
SRT-CLTR	63.36	4.16	3.16	----	15.5
SPART	21.94	13.52	5.52	----	12.7
SDMT	83.24	40.44	11.44	----	37.9
PASAT 3	26.49	8.38	3.38	----	28.4
PASAT 2	33.75	6.56	2.56	----	17.1
SRT-D	8.87	4.28	4.28	----	4.9
SPART-D	9.28	5.18	2.18	----	3.6
WLG	39.88	6.88	0.88	----	17.0
WCST	20	62	101	----	90.50
The Verbal Judgement Task	56	40	18	----	33
Raven’s Standard Progressive Matrices	36	24	10	----	15
HAM-D	6	22	44	----	>10
HAM-A	18	24	30	----	>8

Legend: MMSE = Mini Mental State Examination; FAB = Frontal Assessment battery; SRT-LTS = Selective Reminding Test, Long Term Storage; SRT-CLTR = Selective Reminding Test, Consistent Long-Term Retrieval; SPART = 10/36 Spatial Recall Test; SDMT = Symbol Digit Modalities Test; PASAT 3 = Paced Auditory Serial Addition Test; PASAT 2 = Paced Auditory Serial Addition Test; SRT-D = Delayed Recall of the Selective Reminding Test; SPART-D = Delayed Recall of the 10/36 Spatial Recall Test; WLG = Word List Generation; WCST = Wisconsin Card Sorting Test; HAM-D = Hamilton Depression Inventory Scale; HAM-A = Hamilton Anxiety Rating Scale.

## Data Availability

All data generated or analyzed during this study are included in this published article.

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
