# Peer review of "Neuropsychological Disability in the Case of Natalizumab-Related Progressive Multifocal Leukoencephalopathy"

_medicina, 2022, doi:10.3390/medicina58040551_

Round 1

Reviewer 1 Report

The manuscript by Lo Buono et al. describes the neuropsychological performance of an unfortunately lethal case of progressive multifocal leukoencephalopathy (PML) in a multiple sclerosis patient treated with natalizumab. My comments for further improvement can be found hereafter:

  • What was the treatment of the PML in the patient after the natalizumab was discontinued? Was plasmapheresis utilized? Was any of the mirtazapine, maraviroc, and mefloquine utilized?
  • How many natalizumab infusions did the patient received at the time of PML?
  • Did the patient have a prior history of immunosuppressive therapy?
  • Was the patient JCV-positive at the time of natalizumab initiation and what was the JCV index? Any JCV index timepoints between initiation and PML occurrence? Please comment on the PML risk stratification data available in the literature. https://pubmed.ncbi.nlm.nih.gov/28969984/
  • Were there any attempts at preventing IRIS? Was IRIS diagnosed? Any corticosteroid use?
  • How was written consent taken from the patient that has passed away in 2018 (4 years ago)?
  • Are there any differences in the outcome of PML based on the cognitive measures in NAT-treated MS patients? Is there data on this in the literature? What are the risk factors for unfavorable PML outcome? This should be expanded in the Discussion.

Reviewer 2 Report

I’d like to thank the authors and the Editorial Board for the opportunity to review the article submitted to Medicina. The authors' manuscript refers to a very important topic: the functioning of patients with leukoencephalopathy. I believe that the submitted manuscript does deserve its publication after some minor changes. Below I present my comments on the individual sections of the authors’ work.

Overall: I suggest that the presented manuscript should be evaluated by an English native speaker. In one paragraph authors write. They use present and past sentences to describe the same time period, for example: “At the clinical re-evaluation performed, after 30 days (T2), the patient presented a clinical and neuropsychological worsening. She appears to be alert and cooperative. She carries out simple orders and answers questions appropriately. Conversation was possible with the help of the interlocutor, and spontaneous speech was compromised by aphasic deficits. Oral and written lexical-semantic production was characterized by a deficit in access and phonological structuring.” I believe that the native speaker proofreading will improve the overall tone of the manuscript.

Title: The current manuscript title requires a revision. It suggests that the presented article refers to the literature review which is not the case.  The presented study is not a bibliometric analysis, but only a case study based on selected and limited literature. What is more, the phrase “clinical feature” is not correct. I highly suggest that the authors consult an English native speaker with a background in medical sciences and think about changing the manuscript’s title. I suggest something in line with: Neuropsychological disability in the case of natalizumab-related progressive multifocal leukoencephalopathy

Case presentation: Please insert appropriate reference and a short description of the EDSS 1 score, e.g. (EDSS – 1; No disability, minimal signs in one functional system)[X number]. If possible, please specify which FS was dysfunctional in the studied patient – based on the description, I believe that mild dysfunction of the cerebral functions was the basis of the EDSS diagnosis. While it is obvious to a clinical practitioner or scientific researcher working with patients with MS, the EDSS score of 1 might not be very informative to any other specialists or readers of your article.

Case presentation: authors use the phrase “mental confusion”. Did the authors mean delirium? If yes, I highly suggest that authors refer to it as an “acute confusional state” rather than “mental confusion”. Was it diagnosed by a psychiatrist or neuropsychologist? Based on the authors’ description, the only symptom of the ACS would be confabulation, which doesn’t allow one to make that diagnosis. In my opinion, in the current state, the authors’ description of the patient’s functioning shows her cognitive impairment but not ACS. Please specify and widely describe the sentence in lines 76-77.

Case presentation: please put appropriate references for each of the cut-off criteria threshold values presented in Table 1. Are they based on the questionnaire adaptations in line with the patient’s native language? They are not universal and are different for different cultural conditions. Fortunately, this is not a large limitation of the presented results, as a huge neurological disability in the presented case is clearly visible.

Otherwise, the presented article is of a very high scientific quality. I don’t have any additional comments. I believe this article presents a really interesting case and is important evidence for professionals working with MS patients.

In summary: I do recommend the publication of this manuscript in Medicina after some minor changes.

Reviewer 3 Report

Whilst the submitted paper covers important aspect of MS treatment the discussion could be a bit vaster. However, having a special form of "case report", I think the paper is ready to be published.

Round 2

Reviewer 1 Report

The Authors expand on EDSS scores, which is not relevant to the manuscript.

The Authors did not address the IRIS comment and JCV index classification as mentioned in the previous review cycle.

Author Response

  • The Authors expand on EDSS scores, which is not relevant to the manuscript.

We added a short description of the EDSS score as previously required by another reviewer. However, we have narrowed down the part on EDSS as you suggested.

  • The Authors did not address the IRIS comment and JCV index classification as mentioned in the previous review cycle.

We specified in the manuscript that the patient did not suffer from IRIS. Moreover, we commented on the PML risk stratification as suggested by the reviewer.